# Clinical Utility of Recently Food and Drug Administration-Approved IntelliSep Test (Sepsis Biomarker) for Early Diagnosis of Sepsis: Comparison with Other Biomarkers

**DOI:** 10.3390/jcm13164852

**Published:** 2024-08-16

**Authors:** Nima Sarani, Amitava Dasgupta, Maria Enders, Lauren Rowan, Hanan Elsarraj, Sarah Gralnek, Madison Shay, Lucas R. Lemar, Steven Q. Simpson, Mark T. Cunningham, X. Long Zheng

**Affiliations:** 1Department of Emergency Medicine, The University of Kansas Medical Center, Kansas City, KS 66160, USA; nsarani@kumc.edu (N.S.); menders@kumc.edu (M.E.); lrowan@kumc.edu (L.R.); llemar@kumc.edu (L.R.L.); 2Department of Pathology and Laboratory Medicine, The University of Kansas Medical Center, Kansas City, KS 66160, USA; adasgupta@kumc.edu (A.D.); helsarraj@kumc.edu (H.E.); sgralnek@kumc.edu (S.G.); mshay@kumc.edu (M.S.); mcunningham@everestkc.net (M.T.C.); 3Department of Internal Medicine, The University of Kansas Medical Center, Kansas City, KS 66160, USA; ssimpson3@kumc.edu; 4Institute of Reproductive Medicine and Developmental Sciences, The University of Kansas Medical Center, Kansas City, KS 66160, USA

**Keywords:** sepsis, biomarkers, IntelliSep test, monocyte distribution width, von Willebrand factor, ADAMTS13

## Abstract

**Context:** IntelliSep by Cytovale has received United States (U.S.) Food and Drug Administration (FDA) approval as a sepsis biomarker test. However, the clinical utility of this new test is not assessed in emergency departments. **Objective:** We investigated the clinical utility of this test using 44 patients visiting the emergency department at The University of Kansas Medical Center by comparing it with the monocyte distribution width (MDW) and other biomarkers including the von Willebrand factor (vWF) and ADAMTS13. **Design and Methods:** IntelliSep assesses the cellular host response via deformability cytometry of biophysical leukocyte properties and produces a score (IntelliSep Index; ISI: from 0.1 (lowest risk) to 10 (highest risk). We measured the ISI in 44 patients (19 high probability and 25 low probability of sepsis groups) using EDTA-anticoagulated blood. Left over plasma was used for measuring the plasma von Willebrand factor (vWF) and ADAMTS13 antigen by ELISA assays. The MDW was obtained during routine CBC analysis using a Beckman hematology analyzer. The lactate and high-sensitivity troponin I levels were measured using a Beckman analyzer. Procalcitonin was measured using a Cobas e801 analyzer. **Results:** The median ISI was twofold higher in the high-probability group than in the low-probability group (*p* < 0.01) while the median MDW was 34.5% higher in the high-probability group than in the low-probability group (*p* < 0.01). However, the correlation between the ISI and MDW was only modest (r = 0.66). In addition, significantly higher levels of plasma vWF antigen but lower levels of plasma ADAMTS13 antigen in the high-probability group were found, resulting in significantly higher vWF/ADAMTS13 ratios in the high-probability group than in the low-probability group. **Conclusions:** The new IntelliSep test along with vWF/ADAMTS13 ratios may be useful for the early diagnosis of sepsis in patients visiting the emergency department, which appears to be superior to the traditional marker, MDW.

## 1. Introduction

Sepsis is a medical emergency that describes the body’s systemic immunological response to an infectious process, which may cause end-stage organ dysfunction and death [1]. In 2016, the third international consensus (Sepsis-3) conference defined sepsis as a “life-threatening organ dysfunction caused by a deregulated host response to infection” and septic shock as a “subset of sepsis in which underlying circulatory and cellular/metabolic abnormalities are profound enough to substantially increase mortality” [1]. In 2017, an estimated 48.9 million cases of sepsis were recorded worldwide, and 11.0 million sepsis-related deaths were reported, representing 19.7% of all deaths [2]. According to data from the U.S. Centers for Disease Control & Prevention (CDC), 1,000,000 cases of sepsis occur in the U.S. each year, with 258,000 Americans succumbing annually to this life-threatening systemic inflammatory response to infection. At present, the only diagnostic test that has a confirmed role is pathogen identification by blood culture. Although blood culture to identify the pathogen is the gold standard, it takes days or up to a week to receive a result. Thus, it has no real practical role in the initial diagnostic process when a patient first presents to the emergency department and hospital [3]. Thus, rapid and reliable biomarkers for the early diagnosis of sepsis are needed. 

Blood lactate has been used as a biomarker for sepsis, but it is not specific. The physiological source of blood lactate production during sepsis remains controversial. Several studies have demonstrated that an increased lactate level and lactate clearance are associated with mortality in septic patients [4,5]. The normal blood lactate concentration is about 1.0 mEq/L, while hyperlactemia occurs when the lactate concentration ≥ 1.5  mEq/L. Septic shock is associated with a lactate level above 2.0 mEq/L. However, lactate is not specific for sepsis as it can be elevated in many other pathophysiological conditions. Instead, blood lactate is merely an indicator for altered tissue perfusion [6]. Serum procalcitonin, a polypeptide produced by multiple tissues in the body, is not detectable in healthy individuals but is elevated in response to proinflammatory cytokines, such as interleukin-1, interleukin-6, tumor necrosis factor-α, etc. Serum procalcitonin has been used as a biomarker for sepsis, but it may also be elevated in response to severe bacterial infection [7,8,9,10]. Sepsis involves a pathophysiological process rather than a specific syndrome and thus is too complex to be diagnosed using a single measure. Nevertheless, procalcitonin remains a parameter for diagnosing sepsis [10]. An increase in monocyte distribution width (MDW), which is proprietary to Beckman’s hematology analyzers, has also been shown to be a promising marker in the early detection of sepsis. In one study, the authors determined an MDW value of 23.5 as the cut-off point for achieving the optimal accuracy in sepsis diagnosis, with an area under the receiver operating characteristic (ROC) curve of 0.964, a sensitivity of 0.920, and a specificity of 0.929 [11]. A key advantage of the MDW is its availability in the CBC differential counts, which allows acute care providers to begin suspecting a severe infection when the presence of infection is otherwise not considered.

More recently, point-of-care devices have become available for the early diagnosis of sepsis. IntelliSep is the first U.S. FDA-cleared diagnostic tool for assessing the cellular host response to aid in identifying emergency department patients with sepsis and contribute to rapid life-saving decisions [12,13,14]. IntelliSep can measure the unique biophysical changes in white blood cells (i.e., leukocytes) that occur during sepsis [14]. These structural changes can alert healthcare providers that the patient may develop sepsis because of underlying infections [14]. The results are available within 10 min, and the test can predict if a patient will have a high, medium, or low probability of developing sepsis. 

In this study, we compared the IntelliSep Index (ISI) between emergency room patients with high and low probabilities of sepsis. In addition, we measured the levels of procalcitonin, lactate, von Willebrand factor (vWF), and ADAMTS13, a plasma metalloprotease that cleaves vWF, in these patients. Sepsis can lead to the activation of the vascular endothelium, complement, and coagulation systems. Endothelial vWF and ADAMTS13 are important regulators of hemostasis [15], but their dysregulation during sepsis remains poorly studied. As a result, we also measured the plasma levels of vWF and ADAMTS13 antigen in these patients and correlated these biomarkers with the sepsis score and IntelliSep ISI. Here, we report our findings.

## 2. Materials and Methods

IntelliSep test is a point-of-care test (although the analyzer is relatively large compared to a typical point-of-care device) developed by Cytovale (San Francisco, CA, USA). Cytovale provided the following items free of charge: instrument, test cartridges, and on-site technicians to perform IntelliSep testing. The Cytovale IntelliSep test is a semi-quantitative test using only 100 microliters of whole blood collected in an ethylenediaminetetraacetic acid (EDTA) tube that assesses cellular host response via deformability cytometry of biophysical leukocyte properties. The test is intended for use in conjunction with clinical assessments and laboratory findings to aid in the early detection of sepsis with organ dysfunction to manifest within the first 3 days after testing. It is indicated for use in adult patients with signs and symptoms of infection who present to the emergency department. The IntelliSep test generates an ISI that falls within one of three discrete interpretation bands based on the probability of sepsis with organ dysfunction manifesting within the first three days after testing. Band 1 (ISI 0.1 to 4.9) indicates low probability of sepsis. Band 2 (ISI: 5.0 to 6.2) indicates intermediate probability, while Band 3 (ISI: 6.3 to 10.0) indicates high probability of sepsis. This analyzer does not require calibration by the user.

MDW is a parameter produced by the Beckman hematology analyzer during complete blood count analysis. However, MDW is not reported by the instrument for non-emergency department patients. Lactate was analyzed using a Beckman AU analyzer, while procalcitonin was analyzed using a Cobas e801 analyzer. High-sensitivity troponin I was analyzed using a Beckman DXI analyzer (Brea, CA, USA). 

In this study, 44 patients were included (25 patients with low probability of sepsis and 19 patients with high probability). Criteria for classifying patients into high-risk probability for sepsis and low-risk probability of sepsis are listed in Table 1. After analyzing complete blood count (CBC) in all patients, EDTA whole blood was centrifuged, and plasma was used for determination of vWF antigen (in-house) and ADAMTS13 antigen by enzyme-linked immunosorbent assay (ELISA) developed by R&D Systems (Minneapolis, MN, USA), as previously described [16,17]. 

Lactate and procalcitonin are routinely ordered tests for patients with suspicion of sepsis. In some patients, high-sensitivity troponin I was also ordered due to suspicion of cardiac involvement. 

Plasma vWF and ADAMTS13 antigen: Plasma levels of vWF [18] and ADAMTS13 [17] were assessed by an enzyme-linked immunosorbent assay (ELISA), as previously described. The specimens from 10 normal volunteers were also analyzed for comparison with values observed in both patient groups. This study was approved by the Internal Review Board (IRB) of The University of Kansas Medical Center.

All data are presented as individual values, median, and interquartile range (IQR). Statistical analyses were performed using a non-parametric (Mann–Whitney U test) statistical method. A difference was considered statistically significant only at 95% confidence interval using two-tailed test.

## 3. Results

All patients in this study were ≥21 years old. The demographics of these patients are listed in Table 2. The ISI scores obtained using the IntelliSep test were significantly higher in patients with high suspicion of sepsis compared to those with low suspicion of sepsis. In 19 patients with high suspicion of sepsis, the mean ISI was 7.1 (SD: 1.5, Q1–Q3: 6.0–8.2, *n* = 19), while the median score was 7.4. In contrast, in patients with low suspicion of sepsis, the mean ISI was 3.9 (SD: 1.5, Q1–Q3: 2.9–4.9, *n* = 25), and the median value was 3.7. Therefore, the median ISI in patients with high suspicion of sepsis was twofold higher than the median score in the low-probability group. As expected, such differences were statistically significant according to the Mann–Whitney non-parametric test (*p* < 0.01). 

It is assumed that the presence of Band 3 (ISI: 6.3 to 10.0) indicates a high probability of sepsis. As a result, Band 3 should be present in all patients with a high probability of sepsis. In our patient population of 44, 19 patients had a later confirmed diagnosis of sepsis. There were no discordant specimens in the low-probability group, but there were four discordant specimens in the high-probability group (with a later confirmed diagnosis of sepsis), where ISI varied from 3.3 to 4.9. Based on this observation, the sensitivity of the IntelliSep test for sepsis was 94.7%, the specificity was 96%, the positive predictive value was 93.3%, and the negative predictive value was 95.5%. The distribution of bands in our patient population is given in Figure 1. In four cases with a high probability of sepsis, Band 3 was not observed (Table 3). Interestingly, in case 1, where sepsis was confirmed later, only Band 1 was observed, and lactate was also below 2 mmol/L. However, in three other cases with a high probability of sepsis and Band 1, adjudication showed that none of these three patients had sepsis, although the lactate level was significantly elevated. This indicated that lactate was not a specific marker of sepsis. 

MDW values were also significantly elevated in the high-probability group compared to the low probability group. The mean and median MDW in the high-probability group were 25.6 (SD: 5.2, Q1–Q3: 22.0–28.0, *n* = 18) and 24.4, respectively. In contrast, in the low-probability group, the mean and median MDW were 18.9 (SD: 2.4, Q1–Q3: 17.1–20.6, *n* = 18) and 18.1, respectively. MDW data were not available for two low-probability patients and one high-probability patient. The MDW was significantly increased in the high-probability group according to both the independent t-test and the Mann–Whitney U test. Therefore, the increase in the ISI was more remarkable than the increase in the MDW. We observed only modest linear correlation between the ISI and MDW (r = 0.64), suggesting these two parameters can be used to assess different pathophysiological changes in blood cells. The distribution of the ISI and MDW between the high-probability and low-probability patients is shown in Figure 1. 

vWF is released from activated endothelium [19]. The levels are significantly elevated in sepsis [20,21]. We also observed significantly increased levels of vWF antigen in the high-probability group (mean: 674.3%, SD: 512.6%, median: 544%, range: Q1–Q3: 374.0–860.0%, *n* = 19) compared to the low-probability group (mean: 340.5%, SD: 306.4%, median: 261.0%, Q1–Q3: 131.5–439.0, *n* = 25). These results are shown in Figure 2. The vWF antigen level was significantly higher in the high-probability group than in the low-probability group by Mann–Whitney U test (*p* < 0.003). However, we observed a modest linear correlation coefficient between the ISI score and vWF antigen (r = 0.34). Conversely, ADAMTS13, a plasma metalloprotease that cleaves vWF [15], was lower in the high-probability group (mean: 617.8 ng/mL, SD:136.8 ng/mL, median: 623.2 ng/mL, Q1–Q3: 525.7–689.0 ng/mL, *n* = 19) than in the low-probability group (mean: 717.8, SD: 216.9 ng/mL, median: 685.4 ng/mL, Q1–Q3: 603.3–868.9 ng/mL, *n* = 25); (*p* < 0.05) according to the Mann–Whitney U test. As a result, the vWF/ADAMTS13 ratio was significantly higher in the high-probability group (mean: 11.7, SD: 9.4) than in the low-probability group (mean: 5.7, SD: 7.1); (*p* < 0.0001) according to the Mann–Whitney U test. The mean vWF/ADAMTS13 ratio was twofold higher in the high probability group. 

Serum lactate, procalcitonin, and high-sensitivity (hs) troponin were ordered in 24 patients, 6 patients, and 18 patients, respectively. The mean serum lactate level was 1.8 mmol/L (SD: 1.0, *n* = 19) in the high-probability group and 3.2 mmol/L (SD: 1.8) in the low-probability group. These differences were not significant in our patient group (*p* > 0.05). Interestingly, the median value in the high-probability group was lower than that in the low-probability group (1.5 mmol/L vs. 2.0 mmol/L), indicating that lactate may not be a specific marker of sepsis. In five patients belonging to the high-probability group, the mean procalcitonin was 1.5 ng/mL (range: 0.5–2.4 ng/mL), and, in only one patient belonging to the low-probability group, procalcitonin was 0.06 ng/mL. Interestingly, hs-troponin I was elevated in the high-probability group (mean: 16.7 ng/L, SD: 5.6, range: 14.5–18.5 ng/L) compared to the low-probability group (mean: 6.7 ng/L, SD: 7.0, range: 1.0–10.0 ng/L). The difference was highly statistically significant (*p* < 0.01). 

## 4. Discussion

There are few studies where investigators have reported the utility of the IntelliSep test in the emergency room for assessing sepsis. In one study of 266 patients, those with sepsis had a higher IntelliSep Index (ISI) (median = 6.9; interquartile range, 6.1–7.6) than those adjudicated as not septic (median = 4.7; interquartile range, IQR, 3.7–5.9; *p* < 0.001). Patients with a higher ISI had a higher Sequential Organ Failure Assessment and were more likely to be admitted to the hospital compared with those with a lower ISI (83.6% vs. 48.3%, *p* < 0.001) [12]. O’Neil et al. recently reported the validation of the ISI score in a large patient group (*n* = 572) [14]. Sepsis prevalence was 11.1% in Band 1, 28.1% in Band 2, and 49.4% in Band 3. The positive percent agreement of Band 1 was 81.6%, and the negative percent agreement of Band 3 was 80.7%. The authors concluded that an increasing ISI was associated with an increasing probability of sepsis in patients presenting to the emergency department with suspected infection [14]. Our observation agrees with their conclusion, although we observed better sensitivity and specificity with the ISI for the prediction of sepsis. This may be somewhat biased due to our smaller sample size. A further study with the enrolment of more patients may be needed to confirm the sensitivity and specificity of the IntelliSep test for sepsis prediction. Nevertheless, our study is unique in that we compared the ISI with other markers of sepsis, including the MDW, serum lactate level, and vWF/ADAMTS13 ratio, which provides mechanistic insight into the various biomarkers used for assessing sepsis. 

The utility of the MDW in predicting sepsis has been reported [11,22]. Our study also found a significantly elevated MDW in patients with high probability of sepsis compared to those with a low probability. However, the correlation between the ISI and MDW was only modest (r = 0.64). Interestingly, the correlation between the ISI and MDW in the high-probability group was even lower (r = 0.31), suggesting these two tests are measuring different blood cell properties. 

Our observation of elevated vWF and lower ADAMTS13 in the high-probability patients compared with the low-probability patients is quite interesting. The vWF is a large glycoprotein synthesized within megakaryocytes and endothelial cells, and functions as a mediator of both hemostasis and inflammation [19,23]. In critical illnesses such as trauma, sepsis, and burns, plasma vWF significantly increases and is accompanied by a concomitant decrease in ADAMTS13 (a disintegrin and metalloprotease with thrombospondin type-1 repeats 13) activity [24]. Soluble vWF constantly circulates in the blood stream at concentrations between 8 and 14.0 μg/mL [25]. In one recent study, the authors reported that plasma vWF antigen levels were elevated in patients with sepsis (730 ± 384%) (*p* ≤ 0.0001) relative to patients without sepsis (463 ± 460%, *p* ≤ 0.05) and healthy controls (100 ± 65%) based on a study of 40 patients with sepsis and 40 patients without it [21]. In addition, compared to healthy controls, ADAMTS13 activity was significantly reduced in patients with (47 ± 20%, *p* < 0.0001) and without (75 ± 23%) (*p* < 0.01) sepsis compared with that in healthy controls [21]. Next, they examined the role of the vWF and ADAMTS13 ratios in assessing the likelihood of sepsis. At day 1, the vWF/ADAMTS13 ratio was significantly elevated in both patients with and without sepsis in the ICU relative to the healthy controls [21,26]. In our study, the patients with a high probability of sepsis showed a mean of 674% vWF activity, which is similar to what has been reported in previous studies. Moreover, plasma ADAMTS13 activities were also reported to be significantly reduced in high-probability groups compared to low-probability groups [21,26]. The opposite changes in vWF and ADAMTS13 in sepsis reflect the impact of systemic inflammation and released inflammatory cytokines such as inteleukin-6, interferon-gamma, and tissue necrosis factor (TNF)-alpha that stimulate vWF release but suppress ADAMTS13 synthesis in endothelial cells [27] and/or hepatic stellate cells [28]. 

Newly released vWF multimers can be efficiently regulated by ADAMTS13 protease. Decreased levels of plasma ADAMTS13 are associated with a reduced capacity to regulate vWF multimer size [29]. ADAMTS13 is the only plasma metalloprotease that can efficiently cleave the ultralarge vWF released from the endothelium and megakaryocytes/platelets. As a result, an increase in platelet–vessel wall interactions can lead to thrombotic microangiopathy, clinically manifesting as a syndrome like thrombotic thrombocytopenic purpura (TTP). Patients with TTP may present with severe thrombocytopenia and hemolytic anemia with extremely elevated lactate dehydrogenase, resulting from hemolysis and organ ischemia [30,31]. Sepsis, particularly septic shock, is another clinical condition in which thrombotic microangiopathy may occur, and this is often associated with relatively decreased levels of ADAMTS13 but that are usually greater than 10% of normal [32]. Up to one-third of patients with sepsis have ADAMTS13 levels that are < 50% of normal [32,33]. Our study indicates that the vWF/ADAMTS13 ratio may have potential as a surrogate marker of sepsis. The lack of correlation between the plasma vWF or vWF/ADAMTS13 ratio and IntelliSep results suggests that these two markers probably measure different pathways of activation for sepsis (endothelial activation vs. neutrophil/monocyte activation).

We conclude that this study provides insights into using IntelliSep for early sepsis detection in a clinical setting. The findings indicate that IntelliSep, combined with plasma vWF/ADAMTS13 ratio, holds promise for enhancing early sepsis detection and treatment in emergency scenarios.

## Figures and Tables

**Figure 1 jcm-13-04852-f001:**
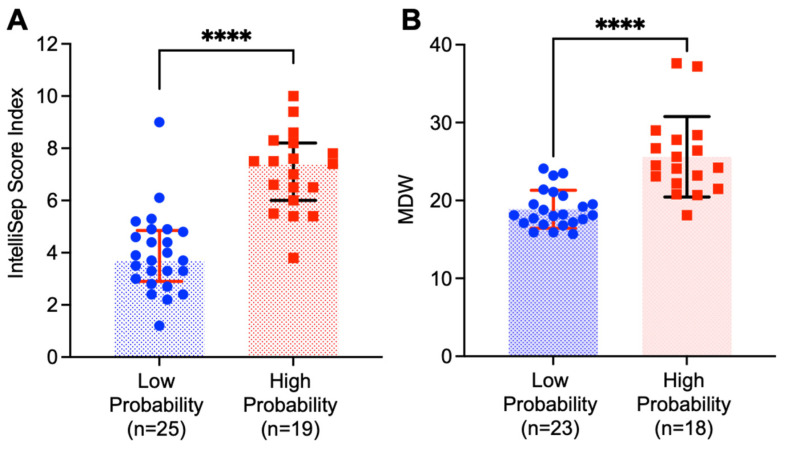
Distribution of IntelliSep Index, ISI (**A**) and monocyte distribution width, MDW (**B**) values among patients with high (pink bars and dots) and low (blue bars and dots) probabilities of sepsis. Data show the individual values (dots), median (bar), and interquartile range (IQR) (horizontal lines). Mann–Whitney U test was performed to determine the statistical significance. Here, **** indicates *p* value less than 0.0001.

**Figure 2 jcm-13-04852-f002:**
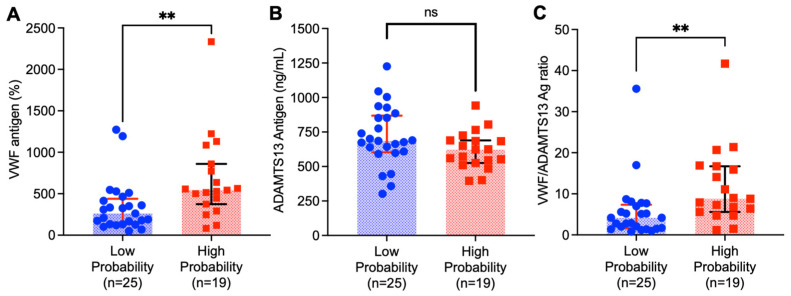
Distribution of plasma von Willebrand factor, vWF (**A**), a disintegrin and metalloprotease with thrombospondin type 1 repeats, 13, ADAMTS13 (**B**), and vWF/ADAMTS13 ratios (**C**) in patients with high (pink bars and dots) and low (blue bars and dots) probabilities of sepsis. Data represent the individual values (dots), median (bar), and interquartile range (IQR) (horizontal lines). Mann–Whitney U test was performed to determine the statistical significance. Here, ns and ** indicate *p* values greater than 0.05 and less than 0.01, respectively.

**Table 1 jcm-13-04852-t001:** Criteria for high probability of sepsis vs. low probability of sepsis.

High Probability of Sepsis	Low Probability of Sepsis
≥18 years old	≥18 years old
Two or more of the systemic inflammatory response syndrome (SIRS) criteria, in which one must be temperature or WBC criterion.	CBC ordered
Prescribed antimicrobials (antibiotics or antivirals).	ED discharged
Admitted to hospital or has an admission order placed.	None

Notes: CBC, complete blood count; ED, emergency department.

**Table 2 jcm-13-04852-t002:** Demographics of patients studied.

Demographic	Count
Female	23
Male	21
Age range (years)	19–97
Asian	1
Black or African American	17
White or Caucasian	24
Other	2

**Table 3 jcm-13-04852-t003:** Four cases where sepsis was confirmed later but Band 3 was not observed.

Case#	Enrolment Criteria	ISI	Interpretation Band (ISI Score)	Comments
1	High probability	3.8	Band 1 (3.8)	Sepsis confirmed, lactate 1.3 mmol/L, MDW: 22.2
2	High probability	3.3	Band 1 (3.3)	No sepsis, lactate: 1.9 mmol/L, MDW: 19.5
3	High probability	4.4	Band 1(3.3)	No sepsis, lactate 6.2 mmol/L
4	High probability	4.9	Band 1 (4.4)	No sepsis, lactate 4.3 mmol/L, MDW 18.0

ISI, IntelliSep Index; MDW, monocyte distribution width.

## Data Availability

Raw and analyzed data are available for sharing upon request from the corresponding author: xzheng2@kumc.edu.

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
