# Peer review of "Clinical Utility of Recently Food and Drug Administration-Approved IntelliSep Test (Sepsis Biomarker) for Early Diagnosis of Sepsis: Comparison with Other Biomarkers"

_jcm, 2024, doi:10.3390/jcm13164852_

Round 1

Reviewer 1 Report

Comments and Suggestions for Authors

The study titled “Clinical Utility of Recently FDA-Approved IntelliSep test (Sepsis biomarker) for Early Diagnosis of Sepsis: Comparison with other Biomarkersreports the clinical utility of a point-of-care device for early diagnosis of sepsis, witch measures unique biophysical changes in the white blood cells. Additionally, others biomarkers were evaluated, including procalcitonin, lactate, endothelial von Willebrand Factor (vWF) and proteolytic enzyme ADAMTS13.

Results demonstrated that the IntelliSep test and the vWF/ADAMTS13 ratio are useful for the early prediction and diagnosis of sepsis. However, several important points need to be reviewed:

-ABSTRACT: clarify acronyms;

-Page 2/ line 59: highlight the importance of procalcitonin as negative predictor marker;

-Page 2/ line 68-69 and 76-78:  insert reference;

-MATERIALS and METHODS:  extensive review needed, starting with a complete description of patients and volunteers (clinical care, sample collection, and processing) and materials used to quantify the different markers;

-Page 3/ line 117-118: insert ethics committee approval;

-Page 3/ line 119-121: review statistical analysis (include normality testes).

-RESULTS: presentation needs to be clearer;

-Table 1: number of patients not related with methodology description;

-Table 2: move to methodology;

-Figure 1: do not report the mean and standard deviation when using a nonparametric statistic;

-DISCUSSION: needs to be expanded.

Comments on the Quality of English Language

not applicable

Author Response

Comment 1. ABSTRACT: clarify acronyms.

Response: Done

Comment 2. Page 2/ line 59: highlight the importance of procalcitonin as negative predictor marker.

Response: We stated in line 99 of revised manuscript that serum procalcitonin is not detected in healthy individuals but significantly elevated in patients with acute inflammation. 

Page 2/ line 68-69 and 76-78:  insert reference. 

Responses: the references have been inserted. 

MATERIALS and METHODS:  extensive review needed, starting with a complete description of patients and volunteers (clinical care, sample collection, and processing) and materials used to quantify the different markers.

Responses: The section has been significantly revised as suggested. 

Page 3/ line 117-118: insert ethics committee approval

Response: IRB approval statement has been added at the end of the manuscript. 

Page 3/ line 119-121: review statistical analysis (include normality testes).

Response: Due to small sample size, we assumed that all data are skewed but not bell-shaped. Thus all analyses were performed using non-parametric statistical tools. This avoids over calling

RESULTS: presentation needs to be clearer. 

Response: We appreciate the comments and we have revised the result section significantly for clarity. 

Table 1: number of patients not related with methodology description. 

Response: The number is corrected due to exclusion of two cases for lack of complete biomarker data. 

Table 2: move to methodology. 

Response: We have changed the Table 2 to Table 1 and moved it to the Method section.

Figure 1: do not report the mean and standard deviation when using a nonparametric statistic

Response: We have changed it to median and interquartile range instead of mean and SD

DISCUSSION: needs to be expanded.

Response: The discussion has been expanded and includes the summary and conclusion at the end. 

Reviewer 2 Report

Comments and Suggestions for Authors

Summary

The research article titled "Clinical Utility of Recently FDA-Approved IntelliSep Test (Sepsis Biomarker) for Early Diagnosis of Sepsis: Comparison with Other Biomarkers" by Nima Sarani et al. investigates the effectiveness of the IntelliSep test in early sepsis diagnosis. This test, developed by Cytovale, measures the biophysical properties of leukocytes to produce an IntelliSep Index (ISI) indicating the probability of sepsis. The study compares the IntelliSep test's performance with other biomarkers such as Monocyte Distribution Width (MDW), von Willebrand factor (vWF), and ADAMTS13 in a cohort of 44 patients from the Emergency Department at the University of Kansas Medical Center.

General Evaluation

The article presents a relevant and timely investigation into a new diagnostic tool for sepsis, an urgent medical condition. The study provides comparative data on the IntelliSep test against established biomarkers, highlighting its potential utility in clinical settings. However, the research lacks clarity in its primary objective and employs a study design that may introduce bias. The statistical analysis is also limited to basic comparisons, which may not adequately support the study's conclusions.

Major Concerns

Clarity of Research Objective:

The research objective is not clearly articulated. It is unclear whether the primary goal is the early detection of sepsis or a comparative analysis of various biomarkers. A precise and well-defined objective is essential to align the study design and methodology appropriately.

Study Design and Bias:

The criteria used to classify patients into high and low probability groups for sepsis introduce potential bias. The study divides patients based on sepsis probability rather than using a more objective criterion such as the presence of SIRS (Systemic Inflammatory Response Syndrome). A more scientifically rigorous approach would be to classify patients based on the actual early diagnosis of sepsis versus those who were not diagnosed early, particularly focusing on patients with SIRS. This approach would provide a clearer assessment of the IntelliSep test's effectiveness in early diagnosis.

Statistical Analysis:

The statistical analysis presented is rudimentary, focusing primarily on basic comparisons between groups. To ensure robust and meaningful conclusions, the study should employ more sophisticated statistical methods. Consulting with a statistician to determine appropriate statistical techniques that align with the research objective is crucial. This might include regression analysis, survival analysis, or other advanced methods suitable for evaluating diagnostic tests.

Author Response

Comment #1.

General Evaluation:  The article presents a relevant and timely investigation into a new diagnostic tool for sepsis, an urgent medical condition. The study provides comparative data on the IntelliSep test against established biomarkers, highlighting its potential utility in clinical settings. However, the research lacks clarity in its primary objective and employs a study design that may introduce bias. The statistical analysis is also limited to basic comparisons, which may not adequately support the study's conclusions.

Responses to Comment #1:  Thank you for your constructive and encouraging comments. We do recognize the limitation of this study. We have selected all patients coming to the ER room for the study. We categorized these patients into high and low probability after the experimental data were collected. Thus, the selection bias is limited. The data structure is quite simple, thus statistical analysis is not complicated involving only basic methology such as student t-test for parametric data and Mann Whitney test for non-parametric data. Due to the low number of subjects, we chose to use the non-parametric test to avoid over interpretation of the data. 

Comment #2.

Major Concerns.  Clarity of Research Objective: The research objective is not clearly articulated. It is unclear whether the primary goal is the early detection of sepsis or a comparative analysis of various biomarkers. A precise and well-defined objective is essential to align the study design and methodology appropriately.

Responses to comment #2: We have restated our study objectives to be more specific as suggested. Our objective is now revised as: We investigated clinical utility of this test using 44 patients visiting the Emergency Department at The University of Kansas Medical Center by comparing it with monocyte distribution width (MDW) and other biomarkers including von Willebrand factor (vWF) and ADAMTS13.

Comment #3.

Study Design and Bias: The criteria used to classify patients into high and low probability groups for sepsis introduce potential bias. The study divides patients based on sepsis probability rather than using a more objective criterion such as the presence of SIRS (Systemic Inflammatory Response Syndrome). A more scientifically rigorous approach would be to classify patients based on the actual early diagnosis of sepsis versus those who were not diagnosed early, particularly focusing on patients with SIRS. This approach would provide a clearer assessment of the IntelliSep test's effectiveness in early diagnosis.

Responses to comment #3: We understood the limitation categorizing the sepsis patients into two groups based on clinical observation, but not objective criteria for the presence of SIRS. However, this is routinely done in our clinical practice and we do not always have the objective data at hand during the ER visit. 

Comment #4

Statistical Analysis: The statistical analysis presented is rudimentary, focusing primarily on basic comparisons between groups. To ensure robust and meaningful conclusions, the study should employ more sophisticated statistical methods. Consulting with a statistician to determine appropriate statistical techniques that align with the research objective is crucial. This might include regression analysis, survival analysis, or other advanced methods suitable for evaluating diagnostic tests.

Responses to comment #4: We understood the limitation for lacking the comprehensive data in term of treatment, mortality, and other outcomes related to the diagnosis of sepsis. However, this will be our future goal for further study. The data we have thus far are simple and do not need complicated statistical analysis. When we have more data and more patients, we do consult our statistician for comprehensive data analysis. 

Reviewer 3 Report

Comments and Suggestions for Authors

The manuscript titled "Clinical Utility of Recently FDA-Approved IntelliSep Test (Sepsis Biomarker) for Early Diagnosis of Sepsis: Comparison with Other Biomarkers" presents comprehensive research into the efficacy of the IntelliSep test compared to other biomarkers for early sepsis diagnosis.

The study's methodology is strong with an organized examination of IntelliSep Index (ISI) values among patients with varying probabilities of sepsis.

The results are encouraging showing that the median IntelliSep Index in the high probability group is notably higher than in the low probability group (p < 0.01). This suggests that IntelliSep may potentially be a diagnostic tool than Monocyte distribution width (MDW) another parameter studied by the researchers, which also displayed a notable increase in the high probability group but had a modest correlation with ISI (r = 0.64). Furthermore, the study delves into Endothelial von Willebrand Factor (vWF) and its proteolytic enzyme ADAMTS13s role in diagnosing sepsis. The researchers noted elevated levels of vWF and decreased levels of ADAMTS13, in the high probability group leading to a higher vWF/ADAMTS13 ratio.

The discovery aligns with studies indicating an imbalance, between these two biomarkers in sepsis caused by inflammation and cytokine release. The research paper effectively links these findings to potential understandings proposing that IntelliSep and the vWF/ADAMTS13 ratio could complement each other in diagnosing sepsis by targeting pathways of activation (endothelial versus leukocyte activation).

A notable strength of this paper is its detailed statistical analysis. By utilizing both parametric tests the study enhances the credibility of its conclusions ensuring they are statistically robust.

However, the study does have some limitations which I suggest to be included to the manuscript text. For instance, the sample size is 44 patients which may impact the generalizability of the results.  Moreover, while the research demonstrates that IntelliSep can deliver results within 10 minutes further exploration is needed to assess how feasible it is to integrate this technology into standard emergency department workflows.

I also recommend that the authors clarify and highlight the conclusions of the study.

In summary this paper provides insights into using IntelliSep for sepsis detection in a clinical setting. The findings indicate that IntelliSep, combined with vWF/ADAMTS13 ratio holds promise for enhancing early sepsis detection and treatment, in emergency scenarios.

I therefore recommend minor revisions addressing all above inconsistencies.

Author Response

Comments:

The manuscript titled "Clinical Utility of Recently FDA-Approved IntelliSep Test (Sepsis Biomarker) for Early Diagnosis of Sepsis: Comparison with Other Biomarkers" presents comprehensive research into the efficacy of the IntelliSep test compared to other biomarkers for early sepsis diagnosis.

The study's methodology is strong with an organized examination of IntelliSep Index (ISI) values among patients with varying probabilities of sepsis.

The results are encouraging showing that the median IntelliSep Index in the high probability group is notably higher than in the low probability group (p < 0.01). This suggests that IntelliSep may potentially be a diagnostic tool than Monocyte distribution width (MDW) another parameter studied by the researchers, which also displayed a notable increase in the high probability group but had a modest correlation with ISI (r = 0.64). Furthermore, the study delves into Endothelial von Willebrand Factor (vWF) and its proteolytic enzyme ADAMTS13s role in diagnosing sepsis. The researchers noted elevated levels of vWF and decreased levels of ADAMTS13, in the high probability group leading to a higher vWF/ADAMTS13 ratio.

The discovery aligns with studies indicating an imbalance, between these two biomarkers in sepsis caused by inflammation and cytokine release. The research paper effectively links these findings to potential understandings proposing that IntelliSep and the vWF/ADAMTS13 ratio could complement each other in diagnosing sepsis by targeting pathways of activation (endothelial versus leukocyte activation).

A notable strength of this paper is its detailed statistical analysis. By utilizing both parametric tests the study enhances the credibility of its conclusions ensuring they are statistically robust.

However, the study does have some limitations which I suggest to be included to the manuscript text. For instance, the sample size is 44 patients which may impact the generalizability of the results.  Moreover, while the research demonstrates that IntelliSep can deliver results within 10 minutes further exploration is needed to assess how feasible it is to integrate this technology into standard emergency department workflows.

I also recommend that the authors clarify and highlight the conclusions of the study.

In summary this paper provides insights into using IntelliSep for sepsis detection in a clinical setting. The findings indicate that IntelliSep, combined with vWF/ADAMTS13 ratio holds promise for enhancing early sepsis detection and treatment, in emergency scenarios.

I therefore recommend minor revisions addressing all above inconsistencies.

Responses: 

We appreciate this reivewer constructive and encouraging comments and support. We have added the summary of our research findings as recommended. 

Round 2

Reviewer 1 Report

Comments and Suggestions for Authors

Author Response

Thank you for your comments. 

Reviewer 2 Report

Comments and Suggestions for Authors

The revised manuscript titled "Clinical Utility of Recently FDA-Approved IntelliSep test (Sepsis biomarker) for Early Diagnosis of Sepsis: Comparison with other Biomarkers" has addressed several important issues; however, some critical methodological concerns remain unresolved. Below are the primary issues that need to be addressed for this study to be considered robust and valid:

  1. Primary Outcome - Sepsis Occurrence: The primary objective of evaluating the diagnostic efficacy of the IntelliSep test in predicting sepsis is not sufficiently met in the current methodology. For a valid assessment of the IntelliSep test’s diagnostic efficacy, it is crucial to stratify the patients based on the actual occurrence of sepsis and determine if the IntelliSep index accurately differentiates between those who develop sepsis and those who do not. This requires a direct comparison of the IntelliSep index against the confirmed diagnosis of sepsis rather than simply comparing the index values across predefined groups.

  2. Scientific Basis for Group Classification: The classification criteria used to define the high and low probability groups for sepsis (Table 1) appear to be subjective and lack a solid scientific foundation. The criteria seem arbitrary and may introduce significant selection bias, which undermines the validity of the study’s findings. The current criteria for group classification do not adequately reflect standard clinical practice or evidence-based thresholds for sepsis prediction, making it difficult to draw reliable conclusions about the IntelliSep test's efficacy.

Methodological Concerns and Recommendations

Given the issues with the current group classification, a more scientifically rigorous approach would be to analyze the correlation between the IntelliSep index and various established sepsis biomarkers directly. A more robust methodology to evaluate the diagnostic efficacy of IntelliSep would involve the following:

  1. Multivariate Logistic Analysis for Sepsis Diagnosis: Conducting a multivariate logistic regression analysis to determine the independent predictive value of the IntelliSep index in diagnosing sepsis compared to other biomarkers. This approach would provide a clear understanding of how well the IntelliSep index performs in conjunction with or independently of other markers. This analysis is feasible with the current dataset and would significantly enhance the study’s validity.

  2. Correlation Analysis: Assessing the correlation between the IntelliSep index and a range of well-established sepsis biomarkers such as procalcitonin, CRP, and WBC counts. Including a larger and more diverse set of biomarkers would provide a comprehensive evaluation of the IntelliSep test. If multivariate analysis is not feasible, this simpler correlation analysis can still provide valuable insights.

  3. Improving Novelty: Although focusing on correlation analysis with known markers may reduce the perceived novelty of the study, it is essential for ensuring methodological soundness and the validity of the conclusions. Including well-established markers such as procalcitonin, CRP, and WBC counts can enhance the robustness of the findings, even if it slightly diminishes the novelty.

Author Response

The revised manuscript titled "Clinical Utility of Recently FDA-Approved IntelliSep test (Sepsis biomarker) for Early Diagnosis of Sepsis: Comparison with other Biomarkers" has addressed several important issues; however, some critical methodological concerns remain unresolved. Below are the primary issues that need to be addressed for this study to be considered robust and valid:

1)  Primary Outcome - Sepsis Occurrence: The primary objective of evaluating the diagnostic efficacy of the IntelliSep test in predicting sepsis is not sufficiently met in the current methodology. For a valid assessment of the IntelliSep test’s diagnostic efficacy, it is crucial to stratify the patients based on the actual occurrence of sepsis and determine if the IntelliSep index accurately differentiates between those who develop sepsis and those who do not. This requires a direct comparison of the IntelliSep index against the confirmed diagnosis of sepsis rather than simply comparing the index values across predefined groups.

All patients with clinical diagnosis of sepsis were confirmed by infectious disease team following admission of these patients to hospital and treated as such for their illness. It is based on a set of clinical criteria for sepsis. There is no single laboratory test that can be used to confirm sepsis. Therefore, our current diagnosis and classification of sepsis are the best practice in the field. 

2) Scientific Basis for Group Classification: The classification criteria used to define the high and low probability groups for sepsis (Table 1) appear to be subjective and lack a solid scientific foundation. The criteria seem arbitrary and may introduce significant selection bias, which undermines the validity of the study’s findings. The current criteria for group classification do not adequately reflect standard clinical practice or evidence-based thresholds for sepsis prediction, making it difficult to draw reliable conclusions about the IntelliSep test's efficacy.

Similar responses to this question. The diagnosis and classification of sepsis are based on a combination of clinical and laboratory information available to date. 

3) Methodological Concerns and Recommendations

Given the issues with the current group classification, a more scientifically rigorous approach would be to analyze the correlation between the IntelliSep index and various established sepsis biomarkers directly. A more robust methodology to evaluate the diagnostic efficacy of IntelliSep would involve the following:

3a) Multivariate Logistic Analysis for Sepsis Diagnosis: Conducting a multivariate logistic regression analysis to determine the independent predictive value of the IntelliSep index in diagnosing sepsis compared to other biomarkers. This approach would provide a clear understanding of how well the IntelliSep index performs in conjunction with or independently of other markers. This analysis is feasible with the current dataset and would significantly enhance the study’s validity.

3b) Correlation Analysis: Assessing the correlation between the IntelliSep index and a range of well-established sepsis biomarkers such as procalcitonin, CRP, and WBC counts. Including a larger and more diverse set of biomarkers would provide a comprehensive evaluation of the IntelliSep test. If multivariate analysis is not feasible, this simpler correlation analysis can still provide valuable insights.

Given the limited number of cases studied, it is not possible to obtain results in multivariate regression analysis or correlation analysis. These are great idea with a large data set we are collecting right now. This will be exceeding our current scope of the study and will not yield any useful data in a year or two unfortunately. This is considered to be a pilot study that may guide our future design a better and more rigorous study in this subject. 

4. Improving Novelty: Although focusing on correlation analysis with known markers may reduce the perceived novelty of the study, it is essential for ensuring methodological soundness and the validity of the conclusions. Including well-established markers such as procalcitonin, CRP, and WBC counts can enhance the robustness of the findings, even if it slightly diminishes the novelty.

We agree by comparison between Instellisep test with many other host response gene products will yield some useful information. However, current data set with a small number of cases will not yield useful information as we have performed some correlation tests. This is by far the first study comparing Intellisep with some of well established biomarkers and novel biomarker ADAMTS13/VWF for sepsis prediction.